# Effect of Hydrophobic Silica Nanochannel Structure on the Running Speed of a Colloidal Damper

**Gengbiao Chen** [1,2,*] **and Zhiwen Liu** [1]

1   College of Automotive and Mechanical Engineering, Changsha University of Science and Technology, Changsha 410114, China; 20203030594@stu.csust.edu.cn
2   Key Laboratory of Safety Design and Reliability Technology for Engineering Vehicle, Changsha 410114, China
*   Correspondence: 006529@csust.edu.cn

**Abstract:** A colloidal damper (CD) can dissipate a significant amount of vibrations and impact energy owing to the interface power that is generated when it is used. It is of great practical significance to study the influence of the nanochannel structure of hydrophobic silica gel in the CD damping medium on the running speed of the CD. The fractal theory was applied to observe the characteristics of the micropore structure of the hydrophobic silica gel by scanning electron microscopy (SEM), the primary particles were selected to carry out fractal analysis, and the two-dimensional fractal dimension of the pore area and the tortuous fractal dimension of the hydrophobic silica gel pore structure were calculated. The fractal percolation model of water in hydrophobic silica nanochannels based on the slip theory could thus be obtained. This model revealed the relationship between the micropore structure parameters of the silica gel and the running speed of the CD. The CD running speed increases with the addition of grafted molecules and the reduction in pore size of the silica gel particles. Continuous loading velocity testing of the CD loaded with hydrophobic silica gels with different pore structures was conducted. By comparing the experimental results with the calculation results of the fractal percolation model, it was determined that the fractal percolation model can better characterize the change trend of the CD running velocity for the first loading, but the fractal dimension was changed from the second loading, caused by the small amount of water retained in the nanochannel, leading to the failure of fractal characterization.

**Keywords:** colloidal damper (CD); hydrophobic silica gel; nanochannel; fractal percolation model; fractal dimension

## 1. Introduction

Vibration and noise reduction are a technical issue of general concern in the field of engineering. Eroshenko proposed a new energy dissipation damping system, called the Colloidal Damper (CD) [1], the structure of which is similar to that of the Hydraulic Damper (HD). The CD uses a colloidal suspension consisting of porous media and liquid as the damping medium (usually porous hydrophobic silica gel and pure water with an antifreeze agent). There are many complex nanopore channels in the porous media, the surface is liquid-sparse, and it does not infiltrate the liquid. In terms of the energy dissipation mechanism, when the liquid enters the nanochannels of the porous media under external pressure, interfacial power is generated at the "gas–liquid–solid" boundary to consume mechanical energy, and the energy conversion of porous medium particles is more obvious after surface modification. Experiments have shown that the energy consumption of a CD is 2 to 3 times that of an HD [2].

The "compression–relaxation" cycle of the CD has the characteristics of partial recovery, where the hysteresis area in the first cycle is larger than that in the following cycles. The energy absorbed in the first cycle is the greatest, and this feature is very suitable for the application of CDs in aircraft landing devices and seismic dampers with high impact energy [3]. At present, the research on CDs is focused on exploring the energy dissipation

mechanism, usually involving experimental studies. Researchers have mainly performed pressurization experiments on hydrophobic silica gels with different modification degrees and pore sizes, allowing the "hysteresis curves" of CDs under different conditions to then be obtained. Additionally, the influencing factors of the energy dissipation capacity of CDs can be explored in this process [4–6]. Iwatsubo et al. [7] studied the effect of different vibration frequencies (0~30 Hz) on the energy dissipation efficiency of CDs by dynamic pressure experiments. The results showed that the CDs had different hysteretic curves under different external vibration frequencies. Suciu et al. [8] established three vehicle suspension system models including CDs and found that the damping characteristics of the CDs varied with the excitation frequency. Suciu et al. [9,10] considered the flow of water in the silica gel nanochannels of CDs as macro-flow and analyzed the seepage through Poiseuille theory. It was determined that the complex pore structure of porous silica gel has a great influence on the seepage velocity of water in the channels.

The working mechanism of a CD involves colloid and interface science. The core scientific problem is to explore the law of liquid seepage in nanochannels. Owing to the complexity and disorder of the porous medium channel structure, there are a number of limitations for the traditional theories and methods. Since the microstructures of porous silica gel have the fractal characteristics of statistical self-similarity in a certain scale, this provides a new method to apply fractal theory and study the transport properties of silica gel nanochannels. Yu and Cheng [11] proposed a fractal seepage model including fractal dimension according to the particle size, porosity, and pore structure of porous media, which was in good agreement with the experimental data. Zhang [12] revised the model from reference [11] and proposed a fractal seepage model suitable for gas in nanochannels.

In this paper, fractal theory was applied to calculate the pore area fractal dimension and tortuosity fractal dimension of a hydrophobic silica gel pore structure, and the fractal seepage model of hydrophobic silica gel nanochannel water was obtained by incorporating slip theory. The variation relationship between the pore structure parameters of the hydrophobic silica gel nanochannels and the running speed of the CD was established. Continuous loading velocity testing of the CD was conducted to verify the effectiveness of the fractal dimension seepage model.

## 2. Structure and Fractal Description of the Silica Gel Nanochannel Structure

### 2.1. Silica Gel Nanochannel Structure and Its Parameters

The basic structure of silica gel is composed of Si-O tetrahedrons stacked by chemical bonds. It has high porosity and large specific surface area. Hydrophobic silica gel is based on silica gel and coupled hydrophobic functional groups (e.g., benzene, alkyl (C4, C8, C18), etc.) with covalent bonds. Alkyl generally uses linear chains of silaneyl, and the chemical molecular expression is Si $(CH_3)_2C_mH_{2m+1}$. Because water molecules are polar molecules, hydrophobic functional groups are generally nonpolar chains, as polar molecules and nonpolar molecules are mutually exclusive. Take the silaneyl studied in this paper as an example. The main structure of the alkyl chain is shown in Figure 1; the longer the alkyl, the better the symmetry, the better the nonpolarity, and the stronger the hydrophobic capacity [13,14].

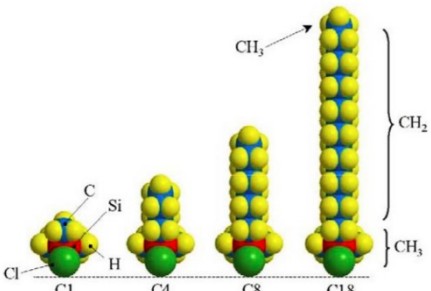

**Figure 1.** Alkyl chlorosilane molecule.

The hydrophobic layer is attached to the surface of the modified silica gel. The thickness of the hydrophobic layer affects the structural parameters of the silica gel particles [9], including pore size distribution, average pore size, specific surface, pore volume, and porosity. The silica gel used in this paper was the Unisil hydrophobic silica gel produced by Suzhou Nano-micro Technology Co, Ltd. (Suzhou, China). Table 1 shows the parameters of the modified silica gel determined and calculated via the $N_2$ adsorption method. In the table, the trade name of the silica gel contains the basic parameters of each silica gel. For example, 10-100 C4 indicates that the particle size of the silica gel is 10 μm, the pore size is 100 Å, and the surface functional group is C4 silica gel. The digit $M$ (4, 8, and 18) in the surface functional group denotes the number of graft molecules of alkyl, and the larger the value of $m$, the stronger the hydrophobicity. The terms $d_a$, $2r_a$, $S_a$, $V_a$, and $\rho_a$ denote the average particle size, average pore size, specific surface area, pore volume, and stacking density of the silica gel particles after modification, respectively. The terms $\lambda_{min}$ and $\lambda_{max}$ denote the minimum and maximum pore sizes of the hydrophobic silica nanochannels, respectively.

**Table 1.** Structural parameters of hydrophobic silica gels.

| Trade Name | 10-100 C4 | 10-100 C8 | 10-100 C18 | 10-200 C18 | 10-300 C18 |
|---|---|---|---|---|---|
| Grafted molecule | C4 | C8 | C18 | C18 | C18 |
| $d_a$ (μm) | 9.42 | 9.42 | 9.51 | 9.68 | 10.20 |
| $2r_a$ (nm) | 10.11 | 10.09 | 10.03 | 20.24 | 30.17 |
| $S_a$ (m$^2 \cdot$g$^{-1}$) | 414 | 414 | 450 | 191 | 113 |
| $V_a$ (mL$\cdot$g$^{-1}$) | 0.95 | 0.95 | 0.98 | 0.90 | 0.91 |
| $\rho_a$ (g$\cdot$mL$^{-1}$) | 0.56 | 0.60 | 0.63 | 0.55 | 0.52 |
| $\lambda_{min}$ (nm) | 3.40 | 2.37 | 1.36 | 4.27 | 8.77 |
| $\lambda_{max}$ (nm) | 20.30 | 21.03 | 22.44 | 35.58 | 51.92 |

Hydrophobic silica gel is a typical granular porous medium with complex pores. Figure 2 shows pore structure images of hydrophobic silica nanochannels with magnifications of 1000 and 100,000, obtained by scanning electron microscopy (SEM). For SEM, we used an FEI Quanta-200; the materials selected were 10-100 C18 and 10-300 C18 silicone, treated with surface gold spray for 15 min. It was determined that the silica gel microspheres (secondary particles) were made up of nanoparticles (primary particles) by necking and bonding. Attributable to the different tightness degrees of each primary particle, there were a number of gaps between the particles, forming nanoscale pore structures with different shapes distributed continuously within the nanoscale. Under multiple magnifications from 1000× to 100,000×, the SEM images of the silicone showed that it is clearly geometric; the structure of silicone at this scale is a fractal structure of self-similarity. The silicone shown in the figure is due to advanced technology, so the shape of the secondary particles is a regular spherical shape, and the particle size distribution of the secondary particles is relatively singular. There is a greater randomness in the process of primary particle aggregation; the cluster size of primary particle aggregation varies, so the aperture distribution between secondary particles is wide [15]. A primary particle of 10-100 C18 silicone was measured using an SEM ruler, and the diameter was about 20–31 nm; for 10-300 C18 silicone, the primary particle size was about 100–110 nm. For these materials, the average apertures measured by the adsorption method passing through $N_2$ were 10.03 nm and 30.17 nm, respectively, so it can be inferred that the size of the silicone primary particles is approximately 2–3 times the average aperture of the silicone; this is roughly in line with the data measured by Yang Junsheng et al. [16]. It was found through the comparison experiments that it was the primary particles, rather than the secondary particles, that affected the percolation and energy dissipation characteristics of the colloidal nanochannels [9]. In this paper, the primary particles were selected for fractal analysis.

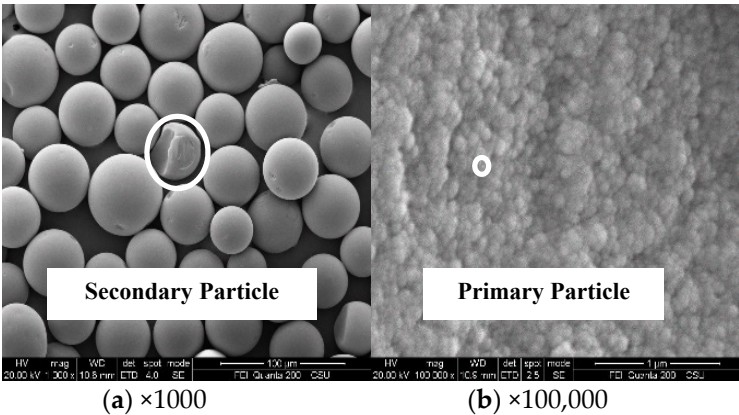

(a) ×1000                                                    (b) ×100,000

**Figure 2.** Pore structure of hydrophobic silica nanochannels.

### 2.2. Fractal Description of Nanochannels

Based on the above analysis, it was assumed that the silica gel nanochannels were composed of a series of independent capillaries, for each of which one end was located on the surface of the particles and the other end was closed inside the particles [17]. The inner surface area of a capillary is equal to the inner surface area of particles, and all flowable spaces are equal to the pore volume of the particles. In practice, a capillary is not a uniform straight circular pipe. It was assumed that each capillary in the porous silica gel had different cross-sectional area and had a certain tortuosity degree $\tau$. In order to characterize the channel structure in the porous silica gel, two fractal characterization parameters were introduced: the pore area fractal dimension $D_f$ and the pore tortuosity fractal dimension $D_t$.

The cross-sectional area and tortuosity degree of capillaries obey the distribution of fractal power law, which was expressed as [12]

$$N(L_0 \geq \lambda_{min}) = \left( \frac{\lambda_{max}}{\lambda} \right)^{D_f} \tag{1}$$

where $N$ is the number of capillaries, $L_0$ is the characteristic length of the fractal capillary set, $\lambda_{min}$ is the minimum diameter of a capillary, and $\lambda_{max}$ is the maximum diameter of a capillary. The values of $\lambda_{min}$ and $\lambda_{max}$ are shown in Table 1.

The expression of the pore area fractal dimension $D_f$ is as follows [10]:

$$D_f = D_E - \frac{\ln \phi}{\ln(\lambda_{min}/\lambda_{max})} \tag{2}$$

where $D_E$ is the Euclidean dimension. $D_E$ is equal to 2 because of the two-dimensional space selected in this paper. $\phi$ is the porous medium's porosity. The prerequisite for the application of fractal theory is that the size distribution of capillaries in the fractal porous medium satisfies the following formula: $(\lambda_{min}/\lambda_{max})^{Df} \cong 0$. The expression of the pore tortuosity fractal dimension $D_t$ is as follows:

$$D_t = 1 + \frac{\ln \tau_{av}}{\ln(L_0/\lambda_{av})} \tag{3}$$

where $\lambda_{av}$ is the average pore size. In this paper, $\lambda_{av}$ equals $2r_a$. $\tau_{av}$ is the average tortuosity of the capillaries. Choosing the appropriate feature length $L_0$ is the key to obtaining $D_t$. Assuming that the fractal capillaries are concentrated on a square with side length $L_0$, $L_0$ can be calculated according to the following formula:

$$L_0 = \sqrt{\frac{(1-\phi)\pi\lambda_{max}^2 D_f}{4\phi\left(2 - D_f\right)}} \tag{4}$$

The average tortuosity in Formula (3) can be calculated by the following formula:

$$\tau_{av} = \frac{1}{2}\left( \frac{\sqrt{1-\phi}}{1-\sqrt{1-\phi}}\sqrt{\left(\frac{1}{\sqrt{1-\phi}}-1\right)^2 + \frac{1}{4}} + \frac{1}{2\sqrt{1-\phi}} + 1 \right) \tag{5}$$

The length of the curved fractal capillary, $L_t$, is as follows:

$$L_t(\lambda) = \lambda_{av}^{(1-D_t)}L_0^{D_t} \tag{6}$$

The average tortuosity values and fractal dimensions of silica gels with different pore structures can be calculated by Formulas (2)–(5). The results are shown in Table 2. It can be seen that the value of $D_f$ ranged from 1.58 to 1.83 and increased with an increase in the number of grafted molecules, whereas it decreased with increasing pore size. The value of $D_t$ ranged from 1.07 to 1.18, showing the opposite trend to $D_f$, decreasing with an increase in the number of grafted molecules and increasing with increasing pore size.

**Table 2.** The fractal dimensions and related parameters of hydrophobic silica gels.

| Trade Name | 10-100 C4 | 10-100 C8 | 10-100 C18 | 10-200 C18 | 10-300 C18 |
|---|---|---|---|---|---|
| $\phi$ | 0.53 | 0.57 | 0.62 | 0.50 | 0.47 |
| $D_f$ | 1.64 | 1.74 | 1.83 | 1.67 | 1.58 |
| $D_t$ | 1.15 | 1.11 | 1.07 | 1.13 | 1.18 |
| $\tau_{av}$ | 1.41 | 1.35 | 1.30 | 1.46 | 1.52 |
| $L_0$ (nm) | 36.43 | 42.11 | 50.97 | 71.33 | 94.13 |

## 3. Fractal Seepage Model

### 3.1. Seepage Theory of Nanochannels

For nanoscale silica nanochannels, the inertia force and volume force in macrohydrodynamics can be neglected, and the surface force dominates [13]. In order to clearly describe the seepage in the silica gel nanochannels, the roughness of the channels was neglected, and they were simplified to a smooth capillary structure. The analysis of gas/liquid seepage in hydrophobic silica nanochannels was performed based on the following assumptions: the interface resistance of the channel was ignored, the two phases of flow in the channel were incompatible with each other, the viscosities of the two-phase fluid did not affect each other, and the temperature did not change during the process of water infiltration [18]. The gas flow in the hydrophobic silica nanochannels was considered to be slip flow and diffusion flow. According to the analysis of gas/liquid two-phase seepage velocity by Zhang [11], it is understood that the gas seepage velocity in hydrophobic silica nanochannels is higher than that of water because of gas diffusion in microflows, so the operation speed of the CD depends on the water seepage velocity in the nanochannels.

The flow pattern of water in nanochannels is an important basis for the study of microfluidic flow. It was assumed that the flow pattern of water in nanochannels was Poiseuille flow and the velocity distribution was parabolic. In Reference [19], the flow in the nanoscale colloidal microporous channel was considered as Poiseuille flow, and then the emergence of the "stagnant ring" was explained by the "Neck-Bottle" theory and the "Contact Angle" theory. Barrat et al. [20] simulated the Couette and Poiseuille flows of liquids at the nanoscale and found the existence of wall "slip" phenomena and density oscillation distributions; Ziarani et al. [21] showed that the velocity distribution was close to the parabolic quadripartite curve, the temperature distribution was close to the quadrangle curve, and the continuous medium theory results were consistent. Figure 3 shows the flow pattern of water in silica gel nanochannels under external pressure $p_m$. Terms $p_\sigma$ and $p_G$ denote the capillary pressure and gas pressure in the nanochannels, respectively. $L$ is the length of the nanochannels, $x$ is the flow distance of water in the nanochannels, and $\Upsilon$ is the surface tension of the solid–liquid interface.

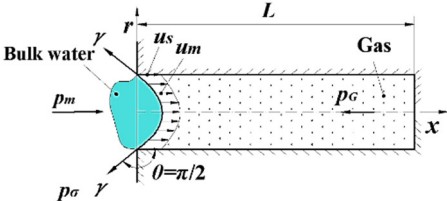

**Figure 3.** Flow patterns of water in hydrophobic silica nanochannels.

In the CD compression stage, when the external pressure $p_m$ is greater than the capillary pressure $p_\sigma$, the bulk water overcomes the capillary pressure $p_\sigma$, gas pressure $p_G$, and other microforces to squeeze air into the nanochannel. In the CD relaxation stage, water seeps through the nanochannel under capillary pressure and gas pressure. The capillary pressure follows the Yang–Laplace formula [8]:

$$P_\sigma = -\frac{4\sigma \cos \theta}{\lambda_{av}} \tag{7}$$

where $\sigma$ is the surface tension of water and $\theta$ is the surface contact angle of the hydrophobic silica gel. Air occupies the silica nanochannels before water infiltration. In this paper, air was considered to be compressible real gas (atmospheric temperature and pressure), without considering gas dissolution in water. The relationship between air pressure and volume can be described by the Van der Waals equation [10]:

$$P_G + \alpha_1 \frac{n}{V_G}(V_G - n\alpha_2) = nRT \tag{8}$$

where $a_1$ is a constant related to intermolecular gravity, 0.1358 $Jm^3 \cdot mol^{-2}$. $a_2$ is a constant related to the volume of the molecule itself, $3.64 \times 10^{-5}$ $m^3 \cdot mol^{-1}$. $n$ is the mole number, $p_G$ is the gas pressure, $V_G$ is the gas volume, and $R$ is the universal gas constant.

For the seepage of water in nanochannels, the continuum theory is no longer applicable, and the molecular discontinuity effect is not particularly obvious in the slip zone, so the slip boundary of the wall should be considered for correction. The slip effect refers to the phenomenon where the velocity of fluid near the wall of the channel/pipeline is not equal to zero. Navier proposed a generalized slip boundary condition, which considered that the slip velocity $u_s$ is proportional to the shear rate of fluid on the wall [22]:

$$u_s = l \frac{\partial u_x}{\partial y}\bigg|_{y=0} \tag{9}$$

where $l$ is the slip length and $l = 0$ is the nonslip boundary condition. The slip length of water in nanochannels is related to the hydrophobicity of the wall—the stronger the hydrophobicity, the longer the slip length of water on the wall. At present, the slip length of smooth hydrophobic surfaces is mainly studied by microscopic experiments. Audry et al. [15] measured the slip length of a smooth Octadecyl trichlorosilane (OTS) surface by using a colloidal probe in Atomic Force Microscopy (AFM), and they found that the slip length of the smooth OTS surface was 6 to 10 nm. The slip velocity $u_s$ was calculated from the slip boundary. Combined with the first-order slip boundary, the seepage velocity of water in a capillary tube can be written as follows [9]:

$$u_m = u_s - \frac{\lambda_{av}^2}{32\mu l}\frac{d_p}{d_x} = \frac{l\lambda_{av}}{4\mu}\frac{d_p}{d_x} - \frac{\lambda_{av}^2}{32\mu l}\frac{d_p}{d_x} \tag{10}$$

where $d_p/d_x$ is the pressure gradient in the nanochannel. Water flow $q$ through the capillary can be calculated based on $u_m$:

$$q(\lambda) = \pi\lambda^2 u_m = \frac{\pi\Delta p}{\mu L_0}\left(\frac{\lambda_{av}^3 l}{16} - \frac{\lambda_{av}^4}{128}\right) \tag{11}$$

where $L$ is the length of the capillary, and $\mu$ is the moving viscosity of water. $\Delta p$ is the pressure difference at both ends of the capillary, and the total pressure difference at both ends of a capillary with length $L$ can be obtained based on the formula $\Delta p = p_m - (p_G + p_\sigma)$.

### 3.2. Fractal Seepage Model of Nanochannels

The pore size of the porous media obeys the fractal power distribution, and the distribution is random. This can be expressed by a probability density function [22].

$$f(\lambda) = D_f \lambda_{max}^{D_f} \lambda_{av}^{-(D_f+1)} \tag{12}$$

Assuming that the porous hydrophobic silica gel nanochannels are composed of curved capillary bundles, as in Reference [11], by combining Formulas (6), (11), and (12), the flow $Q$ through a unit cross section of the capillary can be obtained by integrating a group of fractal capillaries.

$$Q = -\int_{\lambda_{min}}^{\lambda_{max}} q(\lambda)d_{N(\lambda)} = \frac{\pi\Delta p}{128\mu L_0^{D_t}}D_f\lambda_{max}^{D_f}\times$$
$$\left[\left(\frac{8l\lambda_{max}^{D_t-D_f+2}}{D_t-D_f+2} - \frac{\lambda_{max}^{D_t-D_f+3}}{D_t-D_f+3}\right) - \left(\frac{8l\lambda_{min}^{D_t-D_f+2}}{D_t-D_f+2} - \frac{\lambda_{min}^{D_t-D_f+3}}{D_t-D_f+3}\right)\right] \tag{13}$$

In fractal porous media, $(\lambda_{min}/\lambda_{max})^{Df} \cong 0$, $1 < D_f < 2$, and $1 < D_t < 2$, so $(\lambda_{min}/\lambda_{max})^{Dt-Df+3} \cong 0$. Therefore, Formula (13) can be simplified as follows [12]:

$$Q_m = \frac{\pi}{128\mu}\frac{\Delta p D_f}{L_0^{D_t}}\times\left(\frac{8l\lambda_{max}^{D_t+2}-\lambda_{max}^{D_t+2}\lambda_{min}^{D_t-D_f+2}}{D_t-D_f+2} + \frac{\lambda_{max}^{D_t+3}}{D_t-D_f+3}\right) \tag{14}$$

The above formula reveals that the flow rate is related to the channel structure and fractal dimension when silica gel nanochannels are regarded as a group of fractal capillaries. The fractal seepage model can characterize the flow rate in regions with characteristic length $L_0$ under different pore structures when the pressure difference is $\Delta p$. And the total cross-sectional area $A$ of a fractal capillary unit can be expressed as follows [9]:

$$A = \frac{\pi}{4}\frac{D_f}{2-D_f}\frac{1-\phi}{\phi}\lambda_{max}^2 \tag{15}$$

According to the total cross-sectional area $A$, the number of fractal capillary units $n = MV_a/AL_0$ in hydrophobic silica gel with mass $M$ can be calculated. The total flow rate $Q_m$ in hydrophobic silica nanochannels can be obtained by combining with Formula (14):

$$Q_m = \frac{\pi}{128\mu}\frac{MV_a}{A}\frac{\Delta p D_f}{L_0^{D_t+1}}\times\left(\frac{8l\lambda_{max}^{D_t+2}-\lambda_{max}^{D_t+2}\lambda_{min}^{D_t-D_f+2}}{D_t-D_f+2} + \frac{\lambda_{max}^{D_t+3}}{D_t-D_f+3}\right). \tag{16}$$

When the bulk water is compressed, all of it infiltrates the silica gel nanochannels. According to the fact that the macroflow rate in the high-pressure cavity of the CD and the

microflow rate in the nanochannels are conservative, the running speed of the CD, $v_m$, can be obtained [11]:

$$v_m = \frac{4Q_m}{\pi D^2} = \frac{1}{32\mu D^2} \frac{MV_a}{A} \frac{\Delta p D_f}{L_0^{D_t+1}} \times \left( \frac{8l\lambda_{max}^{D_t+2} - \lambda_{max}^{D_t+2}\lambda_{min}^{D_t-D_f+2}}{D_t - D_f + 2} + \frac{\lambda_{max}^{D_t+3}}{D_t - D_f + 3} \right) \quad (17)$$

where $D$ is the piston diameter of the CD. Formula (17) characterizes the relationship between pore structure parameters of the silica gel and the CD operating speed under pressure difference $\Delta p$. Given pressure difference $\Delta p$, the operating speeds of silica gel CDs with different pore structures can be calculated by combining the parameters in Tables 1 and 2. Figure 4 shows the fractal percolation model velocity of 10-100 C$M$ silica gel ($M$ = 4, 8, 18) under constant pressure difference $\Delta p$ ($\Delta p$ = 1 MPa, 2 MPa). Figure 5 shows the fractal percolation model velocity of 10-2$r_a$ C18 silica gel (2$r_a$ = 100 Å, 200 Å, 300 Å) under constant pressure difference $\Delta p$ ($\Delta p$ = 1 MPa, 2 MPa).

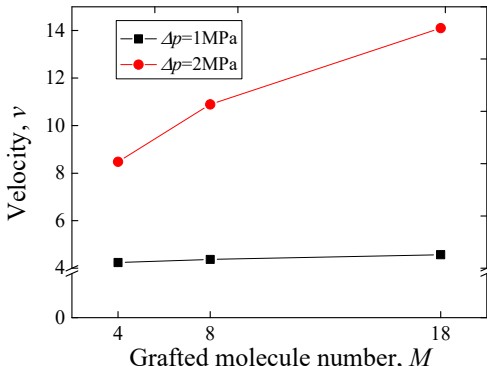

**Figure 4.** Fractal percolation model velocity of 10-100 C$M$ silica gel.

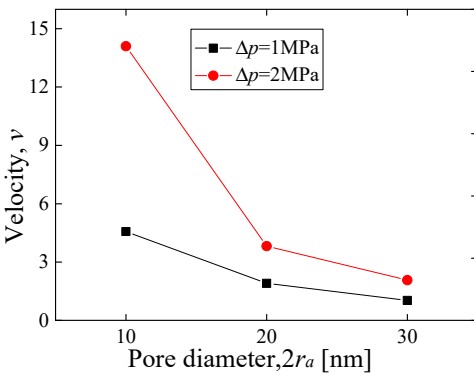

**Figure 5.** Fractal percolation model velocity of 10-2$r_a$ C18 silica gel.

## 4. Experiments and Result Analysis

### 4.1. Experimental Methods

A self-designed CD was adopted. The structure is shown in Figure 6, and it was similar to the single rod piston structure of an HD. However, the main difference between them was that no damping hole or return spring was present in the CD piston. By analyzing the hysteresis curve of the CD, it was determined that the inlet pressure of general silica gel was 10–30 MPa, and when the pressure was greater than 40 MPa, the change in the CD rodless cavity volume was caused by a decrease in the pore diameter of silica gel nanochannels and the volume change when air was compressed [9]. The maximum pressure of the CD high-pressure chamber was 40 MPa. There were five main design indices of the CD: (1) the maximum sealing pressure of the CD rodless chamber was 40 MPa; (2) the diameter of piston D was 30 mm; (3) the maximum velocity of piston $v$ was 6 m·s$^{-1}$; (4) the stroke of

hydraulic cylinder *S* was 150 mm; and (5) the pressure detection port was set to measure the pressure in the rodless chamber.

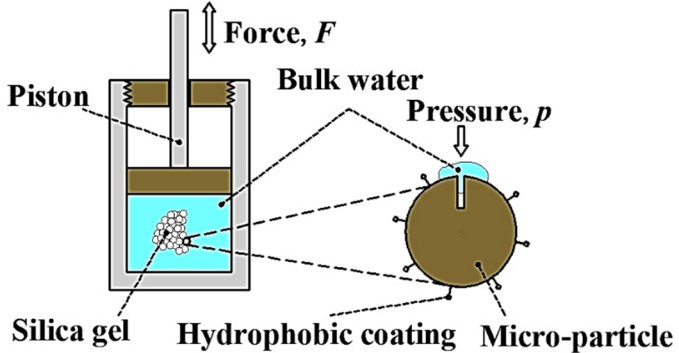

**Figure 6.** Structural sketch of the CD.

The CD speed test bench is shown in Figure 7. A closed-loop electro-hydraulic proportional relief valve controlled the loading pressure, while the loading speed and direction were controlled by an electro-hydraulic proportional direction valve. The CD was fixed on the bench by a fixture. The electro-hydraulic proportional directional valve and the electro-hydraulic proportional relief valve controlled the hydraulic cylinder to load the CD at constant pressure. The pressure and displacement changes in the CD high-pressure chamber were detected by a pressure sensor and displacement sensor, respectively.

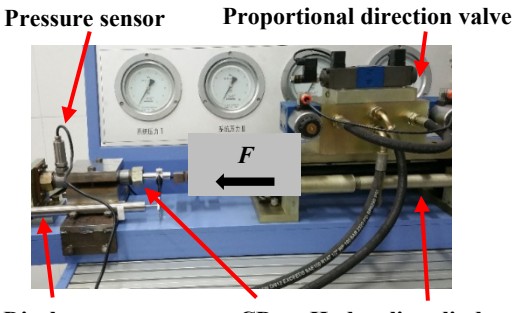

**Figure 7.** Speed test set-up for the CD.

The five types of hydrophobic silica gel shown in Table 1 were used in the experiment. The water used in the experiment was deionized distilled water, and the pH value was 6.0 to 7.0. The amount of silica gel used in each experiment was 4 g, and the volume of pure water used was 40 mL. Continuous loading experiments were carried out on different types of silica gel to test the variation of the CD running speed. During the experiment, the loading speed of the hydraulic cylinder was set to 4 mm·s$^{-1}$, the loading pressure was 40 MPa, and the room temperature was 16 °C. The measured experimental data were the output of the current signal, and the sampling rate of the signal was set to 100. Before each group of experiments, the corresponding pure water compression experiment was carried out, and the experimental data were subtracted from the corresponding pure water compression experiment data to eliminate errors caused by factors such as the cavity deformation and volume change of pure water compression.

## 4.2. Experimental Results and Comparative Analysis

According to the hysteresis of the CD, the bulk phase water did not enter the silica gel nanochannels under low pressure. The water slip flow area was the main energy consumption area of the CD when the external pressure was 2 to 30 MPa [23]. In this paper, the average operating speed of the CD under 2 to 30 MPa pressure was selected

for analysis. Figure 8 presents the experimental values of the running speed of 10-100 C$M$ silica gel CDs ($M$ = 4, 8, 18) under five successive loads. Figure 9 shows the experimental values of the running speed of 10-2$r_a$ C18 silica gel CDs (2$r_a$ = 100 Å, 200 Å, 300 Å) under five successive loads.

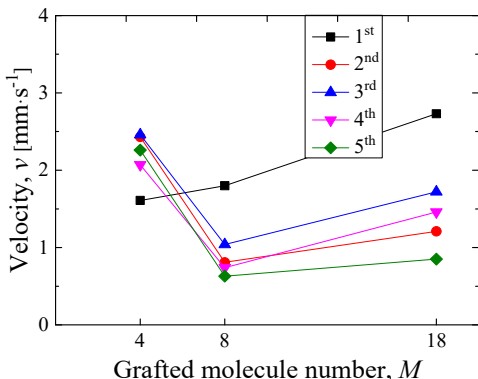

**Figure 8.** Experimental values of the operating speed of 10-100 C$M$ silica gel CDs.

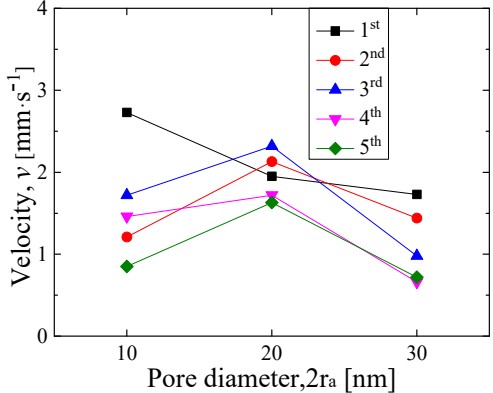

**Figure 9.** Experimental values of the operating speed of 10-2$r_a$ C18 silica gel CDs.

4.2.1. Effect of Number of Functional Groups of Grafted Molecules on the Running Speed of the CD

Figures 4 and 8 show the results of fractal percolation model velocity calculations and the experimental results of CD operation speed for silica gels with different numbers of grafted molecules (10-100 C4, 10-100 C8, and 10-100 C18), respectively.

It can be determined from Figure 4 that the seepage velocity of hydrophobic silica nanochannels was related to the pressure difference: the larger the pressure difference, the faster the seepage velocity. The seepage velocity of nanochannels was also related to the number of grafted molecules on the surface of the silica gel. Theoretically, under the initial conditions of no residual water in the channel, when the pressure difference was constant, there were a greater number of grafted molecules and the seepage velocity was higher. With increasing pressure difference, the difference tended to be more obvious. Owing to the influence of hydrophobic layer thickness, the pore sizes of 10-100 C4, 10-100 C8, and 10-100 C18 decreased; in turn, the hydrophobicity was also enhanced, and the fluid flow velocity was increased [24]. At the same time, because of the different contact angles, the pressure difference when infiltrating nanochannels was also different, which led to the difference in seepage velocity [25].

Comparing Figures 4 and 8, it was found that the fractal seepage model velocity of water in nanochannels under constant pressure difference $\Delta p$ was consistent with that of the CD under initial loading, and it also showed a good fractal characterization effect. However, it was inconsistent with the change trend of the CD running speed after continuous loading.

Relative to those in 10-100 C8 and 10-100 C18, the seepage velocity of water in 10-100 C4 was greater, which is contrary to the theory. This is attributable to the change in the effective pore size of silica gel particles because of the retention of some water in nanochannels after continuous loading. Because the hydrophobicity of 10-100 C8 nanochannels and 10-100 C18 nanochannels is stronger than that of 10-100 C4 nanochannels, the residual water in the 10-100 C4 nanochannels will be relatively small, but under the influence of residual water, the effective pore volume and porosity of the 10-100 C4 nano channels will change more, making the seepage velocity faster.

4.2.2. Effect of Silica Gel Pore Diameter on CD Running Speed

Figures 5 and 9 show the results of fractal percolation model velocity calculations and the experimental results of CD operation speed for different-aperture silica gels (10-100 C18, 10-200 C18, and 10-300 C18).

Figure 4 shows that the seepage velocity of water in nanochannels is related to the pore size of silica gel particles. Under the initial conditions of no residual water in the channel, when the pressure difference was constant, the pore size of the silica gel was larger and the seepage velocity was smaller. This is attributable to the fact that the carbon chain length of 10-100 C18 is 2.45 nm. For the average pore size of 10 nm, the capillary pressure generated by alkane chains can affect the water in the central part of the nanochannel, as well as promoting the water seepage channel after pressure unloading. For 10-200 C18 and 10-300 C18 silica gels with average pore sizes of 20 and 30 nm, respectively, the length of the alkane chain was not enough to generate capillary pressure on the water in the central region of the channel., resulting in the lower seepage velocity of hydrophobic silica gels with larger average pore size after continuous loading.

Comparing Figures 5 and 9, it was found that the fractal seepage velocity of water in nanochannels under a constant pressure difference was consistent with that of the CD under first loading, and it also showed a good fractal characterization effect. However, it was not consistent with the change trend of the CD piston velocity after continuous loading. At this point, with an average pore size of 20.21 nm, there is a significant spike, which differs from the theory. This was also due to the fact that some water was retained in the channel after continuous loading, and the effective pore volume and porosity decreased, resulting in changes in $D_f$ and $D_t$. Because of its nonlinear law of change, the seepage velocity of water in 10-200 C18 channels was the fastest [26].

**5. Conclusions**

In this paper, based on fractal theory, a fractal seepage model reflecting the relationship between the pore structure parameters of porous hydrophobic silica gel nanochannels in CD damped media and changes in the CD operating speed was established. The results of the CD velocity test were compared and verified, and the following conclusions were obtained:

(1)　The primary particles of the hydrophobic silica gel had fractal distribution characteristics. The fractal dimension $D_f$ of the pore area of the hydrophobic silica gel increased with an increase in the number of grafted molecules and decreased with increasing pore size, while the fractal dimension $D_t$ of tortuosity decreased with an increase in the number of grafted molecules and increased with increasing pore size.

(2)　The number of grafted molecules and pore size of the CD hydrophobic silica gel surface functional groups affect the seepage velocity of water in nanochannels. Under the initial conditions of no residual water in the channels, there were more grafted molecules with smaller pore size and a higher seepage velocity, under constant pressure difference.

(3)　The fractal percolation model can effectively characterize the trend of CD velocity change during the first loading, which has great guiding value for the design of CDs for shock absorption. It should be noted that the theoretical average velocity value at the first loading was still less than the experimental value, and the difference was between 11.3% and 24.6%; after continuous loading, $D_f$ and $D_t$ changed because of

the retention of a small amount of water in hydrophobic silica nanochannels, leading to the failure of fractal characterization.

**Author Contributions:** Conceptualization, G.C. and Z.L.; methodology, G.C.; software, Z.L.; data curation, Z.L.; writing—original draft preparation, G.C.; writing—review and editing, Z.L.; visualization, Z.L.; project administration, G.C.; All authors have read and agreed to the published version of the manuscript.

**Funding:** This research was funded by the National Natural Science Foundation of China, grant number 51405036, the scientific research fund of the Hunan provincial education department, grant nummber 15K008, and the Changsha Municipal Natural Science Foundation, grant number kq2014099.

**Institutional Review Board Statement:** Not applicable.

**Informed Consent Statement:** Not applicable.

**Data Availability Statement:** Not applicable.

**Conflicts of Interest:** The authors declare that there are no conflict of interests regarding the publication of this paper.

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
