# Peer review of "Effect of Hydrophobic Silica Nanochannel Structure on the Running Speed of a Colloidal Damper"

_applsci, doi:10.3390/app11156808_

Round 1

Reviewer 1 Report

This article deals with a fractal seepage model for the analysis of the relationship between pore structure parameters of porous hydrophobic silica gel nanochannels in CD damped media and the change of CD operating speed. This article is interesting and suitable for publication in this journal.

The reference should be richer. The abstract reflects well the content of the article however it could be better if authors could appeal to their novelty and future potentials.

As a whole, this article needs a revision before publication.

The major points noted are the following:

  1. Lines 333- 334: “When the pressure difference was constant, there were a greater number of grafted molecules and the seepage velocity was higher.”  This sentence should be better reformulated. In Figure 7, apart from the first load, the velocity of M = 4 is higher than M = 18. Moreover, how the authors explain the minimum value at M = 8?
  2. The authors should better comment on Figure 8 and elucidate the highest values of velocity (apart from the first load) when the pore diameter is 20.24 nm.

Reviewer 2 Report

This important paper can be published after major revisions.
Could you insert an image with a chemical structure of the silica gel?
The SEM measurements just show the results without sufficient discussion and explanations. Provide the full specifications of the SEM instrument used. Specify the parameters applied in SEM micrographs (we can not see them).
Insert references for lines: L42-45.
Kindly, insert a range of values about “...The size distribution of secondary particle of silica gel was narrower, while the size distribution of primary particle was wider.”
Insert more technical details and explain why it was chosen: L177 “....Poiseuille flow and the velocity distribution was parabolic...”
Insert references for all mathematical formulas.
Insert a statistical analysis of the obtained results.
Insert references for Lines 204-205.
Sometimes equations are written aligned to center and sometimes are aligned to right!
L307-308: insert more details about “cavity deformation and volume change of pure water compression”
Minor typos in the manuscript.
Insert Author Contributions.
Minor mistakes in the References (ref. 8).
I recommend to be added to References of the following book:
1. Ţălu Ş., Micro and nanoscale characterization of three dimensional surfaces. Basics and applications. Napoca Star Publishing House, Cluj-Napoca, Romania, 2015.
